# MoAlign: Motion-Centric Representation Alignment for Video Diffusion Models

**Aritra Bhowmik**[*]
University of Amsterdam
a.bhowmik@uva.nl

**Denis Korzhenkov**
Qualcomm AI Research
dkorzhen@qti.qualcomm.com

**Cees G. M. Snoek**
University of Amsterdam
c.g.m.snoek@uva.nl

**Amirhossein Habibian**
Qualcomm AI Research
ahabibia@qti.qualcomm.com

**Mohsen Ghafoorian**[†]
Qualcomm AI Research
mghafoor@qti.qualcomm.com

## Abstract

Text-to-video diffusion models have enabled high-quality video synthesis, yet often fail to generate temporally coherent and physically plausible motion. A key reason is the models' insufficient understanding of complex motions that natural videos often entail. Recent works tackle this problem by aligning diffusion model features with those from pretrained video encoders. However, these encoders mix video appearance and dynamics into entangled features, limiting the benefit of such alignment. In this paper, we propose a motion-centric alignment framework that learns a disentangled motion subspace from a pretrained video encoder. This subspace is optimized to predict ground-truth optical flow, ensuring it captures true motion dynamics. We then align the latent features of a text-to-video diffusion model to this new subspace, enabling the generative model to internalize motion knowledge and generate more plausible videos. Our method improves the physical commonsense in a state-of-the-art video diffusion model, while preserving adherence to textual prompts, as evidenced by empirical evaluations on VideoPhy, VideoPhy2, VBench, and VBench-2.0, along with a user study.

## 1 Introduction

Text-to-video diffusion models have enabled high-fidelity video synthesis across domains from entertainment to simulation. Recent systems like *Wan2.1* (Wan et al., 2025), *CogVideoX* (Yang et al., 2025b), *HunyuanVideo* (Lab, 2025), *PyramidalFlow* (Jin et al., 2025), and *Open-Sora Plan* (Lin et al., 2024a) leverage Diffusion Transformers (DiTs) and large-scale training to achieve impressive visual quality and scalability. Despite high visual quality, these models often generate videos with unnatural motion and physics violations, such as unsupported floating objects, implausible collisions, or inconsistent trajectories. These artifacts reveal a key limitation: while current models excel at generating photorealistic frames, they lack a deep understanding of motion dynamics, which is crucial for producing videos that are both visually and physically plausible.

Efforts to improve the physical plausibility of video generation generally fall into three broad categories: (i) *Simulation-based methods* incorporate physics engines or differentiable simulators in the generation process to model rigid-body dynamics, fluid interactions, or thermodynamic effects (Lin et al., 2024b;c; Liu et al., 2024; Xie et al., 2025; 2024; Zhang et al., 2024; Lin et al., 2025). While effective, these approaches are computationally intensive, domain-specific, and hard to scale to diverse open-world content. (ii) *Non-simulation-based methods* aim to enhance realism without explicit simulation, often by scaling model capacity, leveraging LLM-guided self-refinement, or introducing auxiliary objectives such as 3D point regularization and representation alignment to encourage physically coherent motion (Chen et al., 2025b; Wang et al., 2025; Xue et al., 2025; Zhang et al., 2025b; Hwang et al., 2025). These strategies improve appearance and sometimes temporal consistency, but often prioritize visual semantics over true motion dynamics. (iii) *Conditioning-based*

---

[*]Work was completed while an intern at Qualcomm AI Research

[†]Qualcomm AI Research is an initiative of Qualcomm Technologies, Inc.

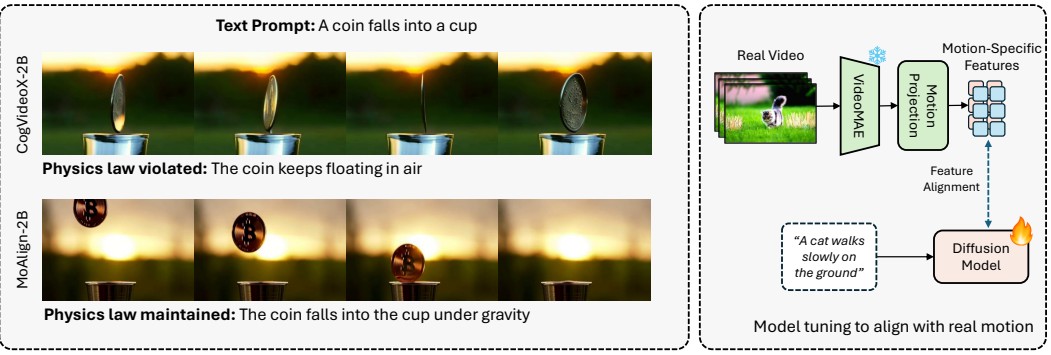

Figure 1: **Problem** *(Left)*: Physics laws are often violated in outputs of video diffusion models. Base CogVideoX model (top) cannot generate a coin falling into a cup: the coin floats in the air instead. Our MoAlign method (bottom) improves this. **Proposed solution** *(Right)*: Our finetuning pipeline aligns internal representations of the diffusion model with motion-specific features extracted from VideoMAEv2.

*approaches* use motion cues like trajectories, optical flow, or pose sequences to guide generation via control mechanisms (Shi et al., 2025; Geng et al., 2025; Zhang et al., 2025c). While effective for temporal coherence, they rely on these extra inputs and preprocessing at inference time, making them impractical for text-only generation.

More broadly, existing methods either rely on external guidance (e.g., physics engines or inference-time controls) or influence motion only indirectly through capacity scaling or appearance-centric alignment, leaving the core issue unresolved: embedding motion understanding directly into the model's latent space. Recent benchmarks reveal that current video diffusion models under-encode motion dynamics in their latent space, leading to identity inconsistency, unstable trajectories, and physics violations, even when individual frames appear realistic (Huang et al., 2024; Zheng et al., 2025; Bansal et al., 2025a). We aim to bridge this gap by aligning the diffusion model's latent features with representations from video encoders trained on real videos, which inherently encode motion observed in the physical world (Zhang et al., 2025b; Hwang et al., 2025). This, however, creates two challenges: (i) alignment may default to matching static appearance features instead of capturing true motion dynamics (Zhang et al., 2025b), and (ii) hard feature matching risks destabilizing pretrained representations during fine-tuning (Hwang et al., 2025). As a result, current alignment-based methods enhance visual fidelity but fall short in enforcing coherent motion. This raises a key question: *how can we design a fine-tuning strategy that explicitly targets motion dynamics, without introducing extra inference-time requirements or compromising model stability?*

To tackle these challenges, we propose a motion-centric fine-tuning framework that disentangles dynamic structure from static appearance. We leverage features from a pretrained video encoder, e.g. VideoMAEv2 (Wang et al., 2023), and learn a projection into a low-dimensional subspace, supervised to predict optical flow, encouraging the subspace to isolate motion-relevant information from entangled semantics. We then align the diffusion model's latent features to these motion representations via a soft relational alignment mechanism. In contrast to prior representation-alignment approaches, e.g. REPA (Leng et al., 2025) or VideoREPA (Zhang et al., 2025b) that relied on joint appearance–motion representations, we make use of motion-only feature space. And unlike Video-JAM (Chefer et al., 2025) which also used optical flows to improve motion coherence, our method does not expand the output space of the diffusion model and does not increase the cost of the inference procedure. We summarize our contributions as follows:

- We suggest a method to learn a motion-specific subspace from a pretrained video encoder by optimizing its projected features to predict ground-true optical flow, enabling a disentangled motion representation.

- We propose to align diffusion model features to this learned motion subspace using soft relational alignment, internalizing motion dynamics without external conditioning or simulation.

- We demonstrate improved temporal coherence and physical plausibility on CogVideoX (Yang et al., 2025b), a state-of-the-art diffusion model, through a user

study and evaluations on physics benchmarks VideoPhy (Bansal et al., 2025a), VideoPhy2 (Bansal et al., 2025b)) while maintaining high visual fidelity in VBench (Huang et al., 2024), and VBench-2.0 (Zheng et al., 2025).

## 2 RELATED WORKS

We group prior efforts to improve physical and temporal realism in text-to-video generation into four areas: architectural advancements, simulation-based methods, conditioning-based motion control, and representation alignment. Each addresses part of the problem, yet none fully internalizes motion dynamics within the generative model as our framework does.

**Text-to-video diffusion models.** Early video diffusion models adapted image pipelines with framewise synthesis (Esser et al., 2023; Geyer et al., 2023; Karjauv et al., 2024) and U-Nets (Ho et al., 2022; Hong et al., 2023; Blattmann et al., 2023b;a; Yahia et al., 2024) but struggled with temporal consistency and realistic motion. Transformer-based designs soon improved spatiotemporal modeling via token compression and self-attention (Villegas et al., 2023; Yan et al., 2021) or linear attention (Chen et al., 2025a; Ghafoorian et al., 2025). Recent systems like CogVideoX, Wan2.1, PyramidalFlow, or Sora push fidelity and scale with 3D-aware representations, pyramidal flow, and spacetime patches, yet videos still show identity drift and physically implausible dynamics (Yang et al., 2025b; Wan et al., 2025; Jin et al., 2025; OpenAI, 2024). We address this gap by internalizing motion through a fine-tuning strategy that disentangles appearance from motion without external conditioning or simulation.

**Simulation-based approaches.** Methods that integrate physics engines or differentiable simulators capture rigid-body, fluid/elastic, or material-aware interactions (Liu et al., 2024; Zhang et al., 2024; Xie et al., 2024; Liu et al., 2025). Some combine simulation with LLM-guided reasoning or handcrafted priors (Xue et al., 2025; Zhang et al., 2025a), and others employ physics-guided generation (Xie et al., 2025; Montanaro et al., 2024). While realism improves, these approaches are domain-specific, compute-heavy, and hard to scale to open-world content. Our method avoids simulation and instead embeds motion understanding directly into the model.

**Condition-based motion control.** Another line conditions generation on motion cues such as optical flow, trajectories, or poses, injected via encoders/adapters (Koroglu et al., 2025; Geng et al., 2025; Terauchi & Yanai, 2021). Plug-and-play customization and temporal in-context fine-tuning further enhance control (Bian et al., 2025; Kim et al., 2025). These methods achieve strong coherence when accurate conditions exist, but require extra inputs or preprocessing at inference, limiting practicality for text-only generation. We instead internalize motion priors within the latent space.

**Representation alignment.** Alignment methods match internal features of generators to pretrained encoders to improve semantics and training efficiency, but are largely spatial and image-centric (Yu et al., 2025; Leng et al., 2025). Video extensions, e.g. VideoREPA, distill spatiotemporal relations via token-level relational matching (Zhang et al., 2025b). However, hard alignment can destabilize pretrained representations and entangled features can mix appearance with motion (Zhang et al., 2025b; Hwang et al., 2025). We build on this direction with *soft relational alignment* to a motion-specific subspace, disentangling dynamics from appearance to internalize motion without sacrificing stability. Besides VideoREPA, closest works to ours are Track4Gen (Jeong et al., 2025) and VideoJAM (Chefer et al., 2025), which introduce motion supervision. Track4Gen operates in an image-to-video setting, using optical-flow-based point trajectories to enforce local correspondence at a single UNet block, but it does not address global motion dynamics or physical plausibility. VideoJAM jointly predicts RGB optical flow, and appearance, injecting motion via inference-time inner-guidance through a learned auxiliary output. In contrast, our method neither predicts flow nor requires inference-time changes: we learn a motion-only subspace from a frozen VideoMAE and align the diffusion transformer to its spatio-temporal geometry, enabling motion priors without altering the generation interface.

## 3 METHOD

Our method builds upon recent advances in video diffusion modeling and representation alignment. We first review the fundamentals of video diffusion models and the REPA framework, which form

the basis of our motion-centric fine-tuning strategy. Then, we introduce our proposed approach for internalizing motion dynamics via soft relational alignment.

## 3.1 PRELIMINARIES

**Video diffusion models.** Modern text-to-video diffusion models, such as CogVideoX (Yang et al., 2025b), generate videos by learning to reverse a forward noising process applied to latent representations of video frames. These models operate in the latent space of a pretrained 3D Variational Autoencoder (VAE), which compresses the input video both spatially and temporally. Let $x_0 \in \mathbb{R}^{F \times H \times W \times C}$ denote a clean video with $F$ frames. The VAE encoder maps $x_0$ to a latent representation $z_0 \in \mathbb{R}^{F' \times H' \times W' \times C'}$, where $F' < F$ due to temporal downsampling. The forward diffusion process perturbs $z_0$ by adding Gaussian noise over $T$ timesteps. The goal of the model is to learn a denoising function $\epsilon_\theta(z_t, t, c)$ that predicts the added noise $\epsilon$, conditioned on the text prompt $c$ and timestep $t$. The training objective minimizes the mean squared error $\mathcal{L}_{\text{diff}}$ between the true and the predicted noise. During inference, the model samples $z_T \sim \mathcal{N}(0, I)$ and iteratively denoises it to obtain $z_0$, which is then decoded by the VAE to produce the final video. CogVideoX employs a transformer-based architecture (MM-DiT) as a denoiser $\epsilon_\theta$. It uses bidirectional spatio-temporal attention to model dependencies across frames.

**Representation alignment.** Diffusion Transformers (DiTs), including those used in CogVideoX, learn internal representations during the denoising process. However, these representations often lag behind those learned by self-supervised visual encoders in terms of semantic richness and discriminative power. REPresentation Alignment (REPA) addresses this gap by introducing a simple yet effective regularization that aligns the hidden states of the diffusion model with pretrained visual features (Yu et al., 2025). Originally, REPA was proposed for image models: Let $x^*$ be a clean input frame and $\mathcal{E}$ a pretrained visual encoder (e.g., DINOv2). The encoder produces a patch-wise representation $\mathbf{Y}^* = \mathcal{E}(x^*) \in \mathbb{R}^{N \times D_e}$, where $N$ is the number of patches and $D_e$ the embedding dimension. During training, the image diffusion model $\mathcal{D}_\xi$ processes a noisy latent input $z_s$ along with condition $c$ and timestep $s$ and produces hidden states $\mathbf{H}_s = \mathcal{D}_\xi(z_s, s, c)$. These are projected via a small trainable network $\mathcal{P}_\phi$ to match the dimensionality of $\mathbf{Y}^*$. REPA encourages alignment by maximizing the similarity between corresponding patches:

$$\mathcal{L}_{\text{REPA}}(\xi, \phi) = -\mathbb{E}_{x^*, \epsilon, s} \left[ \frac{1}{N} \sum_{n=1}^{N} \text{sim} \left( \mathbf{Y}_n^*, \mathcal{P}_\phi(\mathbf{H}_{s,n}) \right) \right], \quad (1)$$

where $\text{sim}(\cdot, \cdot)$ denotes cosine similarity. This loss is added to the standard diffusion objective:

$$\mathcal{L}_{\text{total}} = \mathcal{L}_{\text{diff}} + \lambda \mathcal{L}_{\text{REPA}}, \quad (2)$$

with $\lambda$ controlling the strength of alignment. Empirically, REPA improves convergence speed and generation quality, especially when applied to early transformer layers. However, in case of video models aligning each latent frame independently may lead to temporal inconsistencies, motivating extensions such as cross-frame alignment (Hwang et al., 2025).

## 3.2 MOTION-CENTRIC FINE-TUNING

Our goal is to internalize motion understanding within the diffusion model by aligning its latent features to a *motion-specific subspace*. We achieve this through a two-stage fine-tuning framework: (i) learning motion-centric features from a pretrained video encoder, and (ii) aligning the diffusion model's hidden states to this motion subspace via soft relational alignment. This approach avoids reliance on external simulators or conditioning inputs, and instead distills dynamic structure directly into the generative model.

**Stage 1: Learning motion-centric features.** The objective of this stage is to extract features that encode motion dynamics, disentangled from static appearance and context. This is a challenging task: motion is inherently relational, emerging from temporal changes across frames, whereas appearance is directly observable in individual frames. Consequently, features extracted from pretrained video encoders often entangle motion with appearance, object identity, and scene semantics (Assran et al., 2023; Wang et al., 2021; Zhu et al., 2020). Without explicit supervision, there is no guarantee that learned representations isolate motion-specific information.

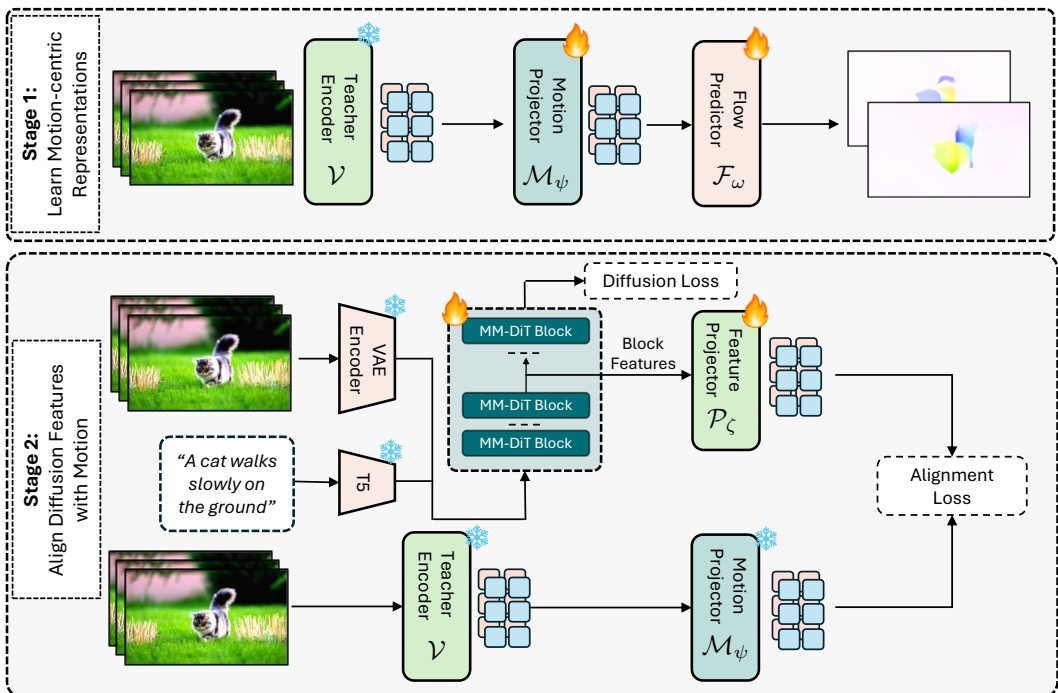

Figure 2: **Overview of our motion-centric fine-tuning framework**. *Stage 1* trains a motion-aware teacher by extracting features from a pretrained video encoder and supervising them with ground-truth optical flow. *Stage 2* aligns the latent features of the video diffusion model (MM-DiT) to the motion-specific subspace via a soft relational alignment loss. This two-stage process internalizes motion understanding without requiring external conditioning or simulation at inference time.

To address this, we *construct a motion-specific subspace* by supervising a projection of pretrained video features to predict optical flow. Given a video clip $x_0 \in \mathbb{R}^{F \times H \times W \times C}$, we extract spatiotemporal features $\mathbf{S} = \mathcal{V}(x_0) \in \mathbb{R}^{F'' \times H'' \times W'' \times D_v}$ using a frozen video encoder $\mathcal{V}$, e.g. VideoMAEv2. These features are projected into a lower-dimensional space via a learnable head $\mathcal{M}_\psi$:

$$\mathbf{M} = \mathcal{M}_\psi(\mathbf{S}) \in \mathbb{R}^{F'' \times H'' \times W'' \times D_m}, \quad D_m \ll D_v. \tag{3}$$

This dimensionality bottleneck is critical. By compressing the feature space, we constrain the model to retain only the most salient information relevant to the downstream task. Prior work has shown that such compression promotes abstraction and suppresses irrelevant appearance cues (Yang et al., 2025a; 2024; Lew et al., 2025). In our case, it biases the representation toward motion by limiting capacity for encoding static content.

To enforce motion specificity, we supervise $\mathbf{M}$ using ground-truth optical flow $\mathcal{O}$ computed between consecutive frames of $x_0$. A lightweight decoder $\mathcal{F}_\omega$ maps $\mathbf{M}$ to predicted flow $\hat{\mathcal{O}} = \mathcal{F}_\omega(\mathbf{M})$, and the training objective is:

$$\mathcal{L}_{\text{flow}}(\psi, \omega) = \left\| \hat{\mathcal{O}} - \mathcal{O} \right\|_1. \tag{4}$$

Optical flow provides dense, low-level supervision that directly encodes pixel-wise motion. By forcing the compressed features to predict flow, we constrain the subspace to encode dynamic structure rather than static semantics. This approach is supported by recent work in motion-aware video modeling, which demonstrates that flow-based supervision improves temporal coherence and physical plausibility (Koroglu et al., 2025; Yang et al., 2024; Lew et al., 2025).

In summary, this stage constructs a motion-specific subspace by (i) compressing high-dimensional video features to suppress appearance, and (ii) enforcing motion supervision via optical flow prediction. The resulting features serve as a distilled representation of dynamics, which we use as a target for aligning the diffusion model in Stage 2.

**Stage 2: Aligning diffusion features to motion.** To internalize motion dynamics within the generative model, we align the latent features of the video diffusion model to the motion-specific subspace learned in Stage 1. We adopt a *soft relational alignment strategy* based on the Token Relation Distillation loss introduced in VideoREPA paper (Zhang et al., 2025b), which matches the pairwise similarity structure of token-level features across space and time.

Consider latent features of the diffusion model $\mathbf{Y}_t \in \mathbb{R}^{\tilde{F} \times \tilde{H} \times \tilde{W} \times \tilde{D}}$ extracted from a noisy input $z_t$. We apply a small projection network $\mathcal{P}_\zeta$ to obtain the tensor $\mathbf{Z} \in \mathbb{R}^{F'' \times H'' \times W'' \times D_m}$ of the same size as $\mathbf{M}$, output of $\mathcal{M}_\psi$. We denote the corresponding spatial features of $f$-th latent frame as $\mathbf{Z}_f$ and $\mathbf{M}_f$, respectively, $1 \leq f \leq F''$. We reshape them to token matrices $\mathbf{Z}_f^\flat, \mathbf{M}_f^\flat \in \mathbb{R}^{H'' \cdot W'' \times D_m}$. The spatial similarity matrix for frame $f$ is defined as:

$$S_Z^{\text{spatial}}(f)[i,j] = \text{sim}(\mathbf{Z}_{f,i}, \mathbf{Z}_{f,j}), \quad S_M^{\text{spatial}}(f)[i,j] = \text{sim}(\mathbf{M}_{f,i}, \mathbf{M}_{f,j}), \tag{5}$$

where $\mathbf{Z}_{f,i}, \mathbf{M}_{f,i} \in \mathbb{R}^{D_m}$ denote the $i$-th token of $\mathbf{Z}_f^\flat$ and $\mathbf{M}_f^\flat$, $1 \leq i, j \leq H'' \cdot W''$. For temporal similarity, we flatten all frames into a sequence of $F'' \cdot H'' \cdot W''$ tokens and compute cross-frame similarities. Let $\mathbf{Z}^{(i)}$ and $\mathbf{M}^{(i)}$ denote the $i$-th token in the full sequence. The temporal similarity matrices are:

$$S_Z^{\text{temporal}}[i,j] = \text{sim}(\mathbf{Z}^{(i)}, \mathbf{Z}^{(j)}), \quad S_M^{\text{temporal}}[i,j] = \text{sim}(\mathbf{M}^{(i)}, \mathbf{M}^{(j)}). \tag{6}$$

As in Sec. 3.1, we employ cosine similarity as the $\text{sim}(\cdot, \cdot)$ function. To emphasize inter-frame dynamics, we exclude intra-frame pairs and apply a temporal weighting scheme. Namely, let $\Delta_{ij}$ denote the distance between frames that tokens with indices $i$ and $j$ belong to, and define the temporal weight matrix $W$:

$$W_{ij} = \begin{cases} \exp\left(-\frac{\Delta_{ij}}{\tau}\right), & \text{if } \Delta_{ij} \neq 0 \\ 0, & \text{otherwise} \end{cases} \tag{7}$$

where $\tau$ is a temperature hyperparameter. The final alignment loss combines spatial and weighted temporal components:

$$\mathcal{L}_{\text{align}}(\theta, \zeta) = \frac{1}{F''} \sum_{f=1}^{F''} \left\| S_Z^{\text{spatial}}(f) - S_M^{\text{spatial}}(f) \right\|_1 + \left\| W \odot S_Z^{\text{temporal}} - W \odot S_M^{\text{temporal}} \right\|_1, \tag{8}$$

where $\odot$ denotes element-wise multiplication and $\|\cdot\|_1$ is the mean absolute error. This formulation extends the original Token Relation Distillation loss by introducing temporal weighting $W$ which prioritizes temporal consistency in the local vicinity of frame. The final training objective equals

$$\mathcal{L}_{\text{total}} = \mathcal{L}_{\text{diff}} + \lambda \mathcal{L}_{\text{align}}, \tag{9}$$

where $\lambda$ controls the strength of motion supervision. This strategy enables the diffusion model to internalize motion dynamics without requiring external conditioning or compromising the stability.

# 4 EXPERIMENTAL SETUP

We detail the implementation of our motion-centric fine-tuning framework, including model architecture, training configurations, and optimization strategies.

## 4.1 MODEL AND TRAINING CONFIGURATION.

We build upon CogVideoX-2B (Yang et al., 2025b), a transformer-based latent video diffusion model composed of MM-DiT blocks with joint spatio-temporal attention. CogVideoX operates in the latent space of a 3D VAE compressing input videos by a factor of 4 along the temporal axis.

**Stage 1: Learning motion-centric features.** For motion supervision, we use Video-MAEv2 (Wang et al., 2023) as a frozen video encoder to extract spatiotemporal features. In Stage 1, these features are compressed using a 3D convolutional network that reduces the channel dimension from 768 to 64 while preserving temporal structure, encouraging the retention of motion-relevant information. The compressed features are then decoded into dense optical flow using a lightweight

Table 1: **VideoPhy2 results.** We report semantic adherence (SA) and physical correctness (PC). Our model achieves highest joint score and demonstrates better trade-off than alternatives.

| Method | SA | PC | Joint |
|---|---|---|---|
| CogVideoX-2B | 27.1 | 64.5 | 22.3 |
| Static baseline | 15.6 | **91.0** | 15.1 |
| CogVideoX-2B (FT) | 26.4 | 73.1 | 22.8 |
| VideoREPA-2B (paper) | 21.0 | 72.5 | – |
| VideoREPA-2B (reimpl.) | 26.1 | 73.3 | 23.0 |
| MoAlign-2B (ours) | **28.8** | 75.0 | **24.9** |

Table 2: **VideoPhy results.** While fine-tuning on our data lowers SA across models, our method maintains a competitive SA and achieves the highest PC scores across all four interaction types, demonstrating robust physical modeling through motion-centric alignment.

| Method | Solid–Solid | | Solid–Fluid | | Fluid–Fluid | | Overall | |
|---|---|---|---|---|---|---|---|---|
| | SA | PC | SA | PC | SA | PC | SA | PC |
| CogVideoX-2B | **24.7** | 16.9 | **67.5** | 24.8 | **69.0** | 40.0 | **49.8** | 23.9 |
| CogVideoX-2B (FT) | 22.5 | 29.6 | 62.1 | 34.5 | 58.2 | 45.5 | 44.9 | 34.1 |
| VideoREPA-2B (reimpl.) | 23.2 | 31.0 | 66.9 | 39.3 | 54.6 | 52.7 | 46.7 | 37.9 |
| MoAlign-2B (ours) | **24.7** | **31.7** | 66.9 | **40.7** | 67.3 | **56.4** | 49.3 | **39.4** |

transposed convolutional network that progressively upsamples spatial resolution in a UNet-like fashion. This setup ensures that the compressed features capture dynamic structure, as they are explicitly trained to regress RAFT-computed ground-truth flow using L1 loss. All VideoMAEv2 weights remain frozen during this stage. We train this stage using the AdamW optimizer with a learning rate of $1 \times 10^{-4}$, $\beta_1$=0.9, $\beta_2$=0.95, and weight decay of $1 \times 10^{-3}$. Training is conducted for 50,000 iterations using four NVIDIA H100 GPUs (80GB VRAM each) with a batch size of 128.

**Stage 2: Aligning diffusion features to motion.** In Stage 2, we align the latent features of CogVideoX to the motion-specific subspace learned in Stage 1. We use a lightweight MLP projector that maps high-dimensional MM-DiT features (1920 channels) to a compact 64-dimensional space via a 4-layer MLP with SiLU activations. The projected features are temporally upsampled and spatially downsampled using a convolutional head. This transformation ensures compatibility with the motion subspace dimensions while preserving relational structure. The alignment is applied to the 18th MM-DiT layer, and optimized using our soft relational alignment loss with temporal weighting. We set $\lambda$=0.5 and $\tau$=10.0, and train using AdamW with a learning rate of $2 \times 10^{-6}$ and batch size 32. Training is conducted for 4000 iterations using four NVIDIA H100 GPUs (80GB VRAM each). We use the AdamW optimizer with a learning rate of $2 \times 10^{-6}$, a batch size of 32, and enable mixed precision training via PyTorch AMP.

**Dataset.** For fine-tuning our models we used a 350K subset of the video dataset used by Open-Sora Plan (Lin et al., 2024a), along with a set of 16K synthetic video samples generated by the Wan2.1 14B model, with prompts sourced from the same set as for the Open-Sora Plan dataset.

## 5 RESULTS

### 5.1 COMPARISON WITH BASELINES

We evaluate our method across three complementary axes of video generation: (i) physical plausibility, using the VideoPhy (Bansal et al., 2025a) and VideoPhy2 (Bansal et al., 2025b) benchmarks; (ii) general generation quality, using VBench (Huang et al., 2024) and VBench-2.0 (Zheng et al., 2025); and (iii) perceptual realism, via a blind user study. Each evaluation targets a distinct aspect of generative fidelity, from adherence to physical laws to semantic alignment and human preference.

**VideoPhy2.** This recent benchmark evaluates physical plausibility while focusing on action-centric scenarios involving human–object interactions. Videos are generated from 591 extended prompts, and scored using the VideoPhy2-AutoEval model. This model predicts two metrics on a 5-point scale: *Semantic Adherence (SA)* which measures how well the video matches the prompt and *Physical Commonsense (PC)* which assesses whether the motion and interactions are physically plausible. The primary metric is the *Joint score*, defined as the fraction of videos rated $\geq 4$ on both SA and PC dimensions. Tab. 1 highlights the importance of holistic evaluation: a degenerate static baseline, which simply repeats the first frame, achieves a deceptively high PC score by avoiding motion violations, but fails on SA, resulting in a low Joint score.

We compare our MoAlign method against the base model CogVideoX-2B and recent VideoREPA approach that aimed to improve physical plausibility. To decouple the effect of training data and different alignment methods, we also finetuned the base model on the same dataset and with the

Table 3: **General video quality.** All methods maintain the original technical quality, as indicated by VBench. On VBench-2.0 MoAlign demonstrates improvement in Total score, mainly driven by improved instance preservation, dynamic spatial relationship and human anatomy.

| Model | VBench | | | VBench-2.0 | | | | | |
|---|---|---|---|---|---|---|---|---|---|
| | Total | Quality | Semantic | Total | Creativity | Commonsense | Controllability | Human Fidelity | Physics |
| CogVideoX-2B | 80.6 | 81.6 | 76.6 | 54.9 | 52.8 | 60.2 | **26.6** | 81.1 | **53.9** |
| CogVideoX-2B (FT) | 80.3 | 81.1 | 77.1 | 54.7 | **58.7** | 60.8 | 25.6 | 83.4 | 44.9 |
| VideoREPA-2B | 80.5 | 81.3 | 77.2 | 55.0 | 56.9 | 61.4 | 25.9 | 85.4 | 45.1 |
| MoAlign-2B (ours) | **81.3** | **82.0** | **78.2** | **55.9** | 52.8 | **65.5** | 25.7 | **86.7** | 48.8 |

same training budget, this checkpoint is referred to as *FT*. As shown in Tab. 1, our training data has marginal effect on Joint score.

Compared to the base model, MoAlign-2B achieves improvement both for individual dimensions and Joint score. Since Zhang et al. (2025b) have not reported the Joint score for their VideoREPA method, we reimplemented it and evaluated independently in the same setup. While VideoREPA-2B improves PC, it suffers from a noticeable drop in SA, resulting in a lower gain in Joint score than our method. This suggests that alignment to entangled features may improve physical realism at the cost of prompt fidelity. In contrast, our method aligns to disentangled motion features, improving both motion realism and semantic alignment. This indicates that internalizing dynamic structure via motion-specific supervision leads to more coherent and faithful video generation. We have further investigated the effectiveness of MoAlign on Wan2.1 (1.3B); the results are available on Appendix A.5

**VideoPhy.** The second benchmark focuses on material-centric interactions across three categories: solid–solid, solid–fluid, and fluid–fluid. Videos are generated from 343 prompts. Scoring is performed using the VideoConPhysics auto-rater, which evaluates SA and PC dimensions. In contrast with VideoPhy2, extended prompts were not standardized in this benchmark. Therefore we opted for sampling videos from all models with the short prompts provided by Bansal et al. (2025a).

As shown in Tab. 2, we observe a consistent trend across all method: finetuning on our data tends to reduce SA scores compared to the base model (note that *FT* model without any alignment performs the worst). We attribute this to the shortage of relevant examples in the dataset. At the same time, PC reflects the plausibility of generated physics irrespective of the fact if it follows the textual prompt. Notably, our method most effectively mitigates the drop in SA among all finetuned variants while achieving the highest PC scores across all interaction types. This proves that the proposed MoAlign training strategy overcomes the limitations of training data better than other considered methods.

**VBench and VBench-2.0.** To ensure that improvements in physical plausibility do not come at the cost of overall video quality, we evaluate all methods with two other commonly used toolkits. VBench focuses more on perceptual characteristics such as aesthetics, temporal smoothness, object-scene consistency, etc., while its second version targets intrinsic faithfulness of generated videos.

We report the aggregated metrics for both benchmarks in Tab. 3. Please refer to the Supplementary for more fine-grained results. First of all, we note that all methods keep the VBench Total score approximately constant which suggests that none of them worsens the technical quality of generations. For VBench-2.0, VideoREPA is on par with the base model in terms of Total score, while our method brings noticeable improvement. This gain is achieved mainly by means of Commonsense and Human Fidelity metrics which cover, among others, such dimensions as instance preservation, dynamic spatial relationship, and human anatomy – highly important aspects for physical plausibility. The significant drop of Physics score for all funetuned models has the same explanation as in VideoPhy case: our training data lacks samples related to thermotics and materials which are pivotal for this category.

**User study.** To complement automated metrics, we conduct a blind user study to assess temporal coherence and physical plausibility from a human perspective. We compare three models: the base CogVideoX-2B, VideoREPA-2B, and our MoAlign-2B. For each model, we generate 50 videos using extended prompts sampled from a mix of VBench-2.0 and VideoPhy2, and collect 672 pairwise preferences in side-by-side comparisons.

Table 4: **Ablation study.** Both proposed components are important for the performance of our method, as measured by VideoPhy2.

| Method | SA | PC | Joint |
|---|---|---|---|
| REPA loss | 25.7 | 71.9 | 22.3 |
| CogVideoX (FT) | 26.4 | 73.1 | 22.8 |
| VideoREPA | 26.1 | 73.3 | 23.0 |
| MoAlign w/o motion features | 27.8 | 73.8 | 23.5 |
| MoAlign w/o soft-TRD loss | 28.2 | 74.4 | 24.1 |
| MoAlign (ours) | **28.8** | **75.0** | **24.9** |

Table 5: **Layer for alignment.** Impact of the feature alignment layer on model quality based on VidePhy2 scores.

| Layer | SA | PC | Joint |
|---|---|---|---|
| 10 | 27.3 | 71.9 | 23.2 |
| 12 | 27.6 | 71.7 | 22.9 |
| 14 | 27.1 | 71.6 | 23.1 |
| 16 | 28.2 | 73.9 | 24.2 |
| 18 | **28.8** | **75.0** | **24.9** |
| 20 | 27.1 | 73.2 | 23.8 |
| 22 | 27.6 | 72.8 | 23.4 |

Table 6: **User study.** Our results are preferred over those from the baselines.

| Comparison | MoAlign | Baseline |
|---|---|---|
| vs. CogVideoX-2B | **68%** | 32% |
| vs. VideoREPA-2B | **78%** | 22% |

As shown in Tab. 6, our method is preferred significantly more often in both comparisons, indicating that motion-centric alignment not only improves physical plausibility and automated scores, but also enhances perceived realism and prompt fidelity. These results confirm that our framework leads to more coherent and visually compelling video generation, as validated by human judgment.

We provide examples of videos used for the user study in the Supplementary.

## 5.2 Ablation Study

We also conducted experiments to isolate the contributions of our method. First, we considered a modification of MoAlign that employs untouched feature extractor from VideoMAEv2 for supervision. Second, we try vanilla TRD loss without weighting based on distance between frames, i.e. $\tau \to \infty$ in Eq. 7. Note that these two ablations done simultaneously result in the VideoREPA method. Additionally, we report the performance of classical REPA alignment, which pushes internal diffusion features towards those from VideoMAEv2 instead of distilling their self-relational structure. Following Zhang et al. (2025b), we use VideoPhy2 for the evaluation of ablated models. Results are reported in Tab. 4. As shown, both our proposed components are important for the performance, and both of them individually improve the VideoREPA baseline. Notably, REPA loss provides the worst results and cannot even surpass simple finetuning of the diffusion model on our data.

Prior works noted the sensitivity of alignment results to the selection of the layer from denoising network $\epsilon_\theta$ for extraction of latent features (Leng et al., 2025; Zhang et al., 2025b). Therefore, we also examined several layers and chose the 18-th transformer block of CogVideoX for our MoAlign method, see Tab. 5. This choice is consistent with REPA-style approaches, which typically align at a single mid-level layer rather than across the entire network. Empirically, we observe that aligning at layer 18 yields the best VideoPhy2 performance, while both earlier and deeper layers lead to lower physical commonsense scores. We also found that distributing the loss across multiple layers slightly degrades results, suggesting that overly broad regularization disrupts the denoising trajectory. These findings indicate that motion-related relational structure is naturally concentrated around mid-depth blocks, motivating our focus on a single alignment layer. More details provided in Appendix A.4.

We also conducted additional analysis on the Stage-1 training dynamics, including convergence behavior and training cost. The full details of this study are provided in Appendix A.3.

## 5.3 Qualitative Results

We show a qualitative comparison in Figure 3. Our MoAlign accurately depicts both the source and target glasses throughout the video, with a clear and coherent pouring motion. In contrast, CogVideoX fails to represent the scene structure, omitting key elements like the second glass and the liquid flow. VideoREPA partially captures the target glass but lacks temporal consistency in the pouring dynamics. This example highlights our model's ability to preserve scene semantics and physical realism. Additional qualitative comparisons are provided in appendix A.1

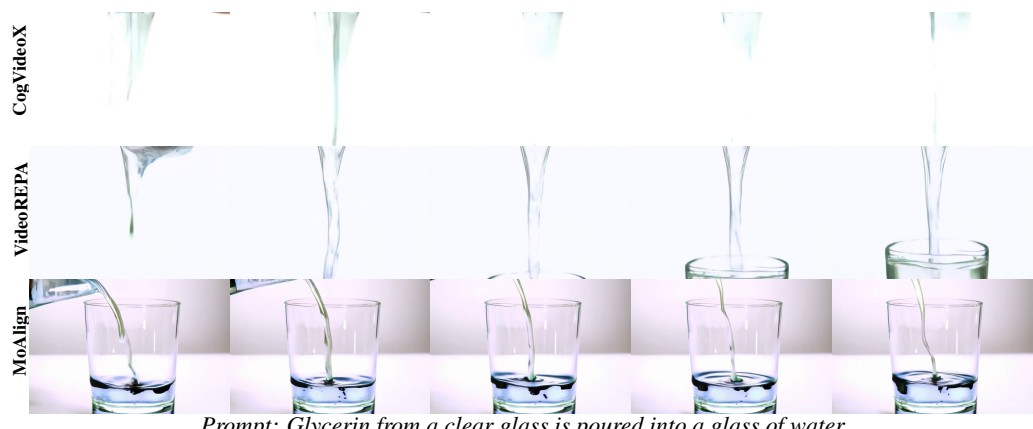

*Prompt: Glycerin from a clear glass is poured into a glass of water*

Figure 3: **Qualitative results.** MoAlign shows both glasses and a coherent pouring motion, while baselines miss key elements or exhibit inconsistent fluid behavior.

## 6   LIMITATIONS

While MoALign consistently improves motion plausibility and physics-centric behavior, it inherits two important limitations. First, its performance is bounded by the coverage of available training data: as noted in Tab. 3, the VBench-2.0 Physics category includes domains such as thermotics, material deformation, liquids, and granular media, none of which appears in the datasets used for our Stage-1 motion-teacher training, or Stage-2 diffusion finetuning. Second, MoAlign provides implicit physics through motion statistics learned from real videos, but does not explicitly model forces, material properties, or long-horizon causal reasoning, and may therefore struggle with scenarios requiring reasoning beyond motion alone, a limitation also discussed in prior works like VideoJAM (Chefer et al., 2025). While addressing these limitations primarily requires video data containing such phenomena, incorporating them would only require retraining the lightweight Stage-1 module followed by standard Stage-2 alignment, without re-training VideoMAE or CogVideoX from scratch. We view extending MoAlign toward richer physics domains as an exciting direction for future work.

## 7   CONCLUSION AND FUTURE WORK

In this work we presented a method for improving temporal coherence and physical plausibility in pretrained video diffusion models. Our pipeline called MoAlign is based on the trainable alignment of internal diffusion features to motion-specific representations extracted from original videos. We demonstrated that such representations result in better quality than general features extracted from a pretrained self-supervised video encoder. Also, we showed that such alignment works better if it prioritizes local vicinity of each frame over long-range temporal dependencies. We evaluated our approach with four recent and commonly used benchmarks, as well as a user preference study.

In our experiments we found that several quality metrics were hurt for all finetuning-based methods that we tried. We attribute that to the limitations of our training dataset, and suggest that MoAlign may further benefit from better data curation. Nevertheless, our method demonstrated stronger resilience to shortcomings of the collected dataset than the baselines.

As a downside of our method, we noted that sometimes it improves the physical commonsense at the expense of reduced motion in the generated videos. We consider this limitation as a viable direction for future work.

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

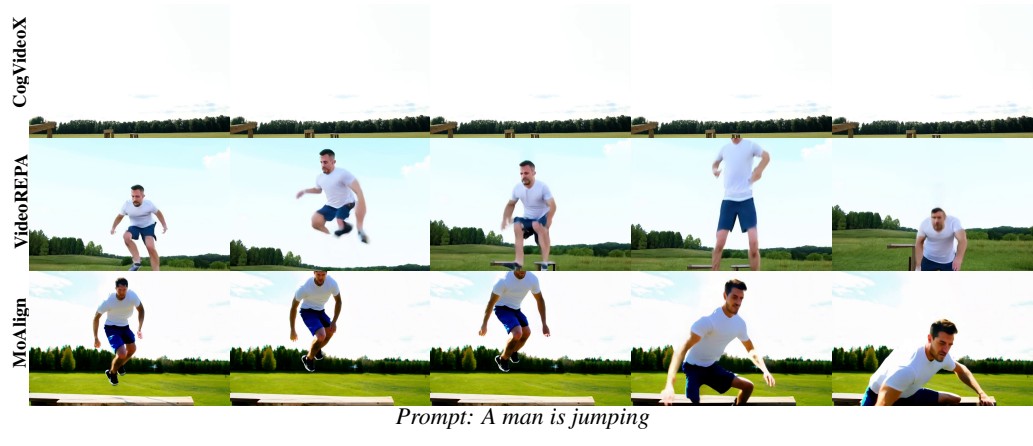

*Prompt: A man is jumping*

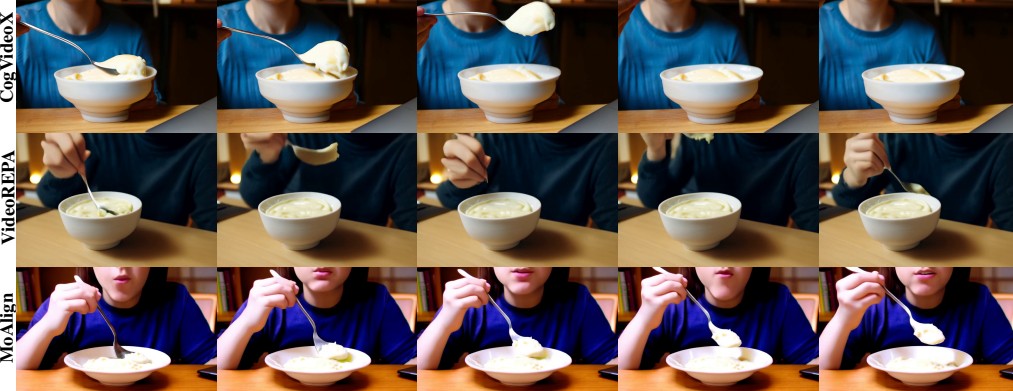

*Prompt: A person is eating pudding with a spoon*

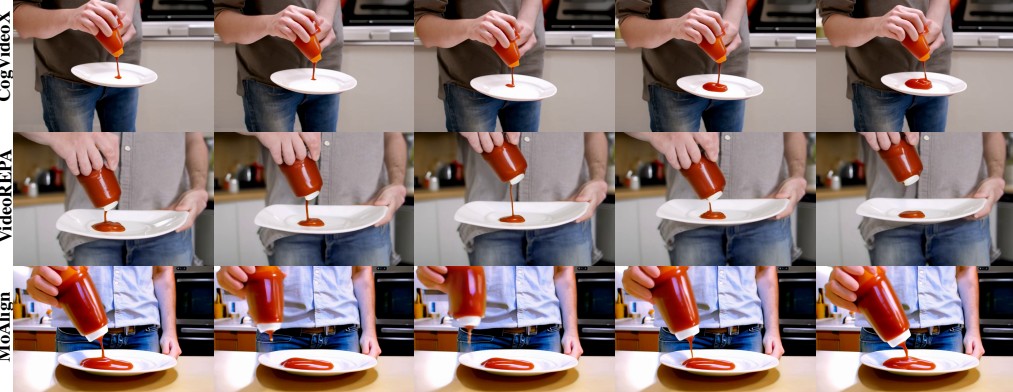

*Prompt: A person is squeezing ketchup onto a plate*

Figure A.1: Qualitative comparison across methods for three prompts. In the first video, our method (MoAlign) preserves realistic human motion without deformation. In the second, it captures accurate hand-mouth interaction while baselines fail to represent the subject. In the third, it models physically plausible ketchup flow, unlike the erratic behavior seen in baselines. See supplementary videos for full comparisons.

## A  APPENDIX

The Appendix consists of the following sections: Qualitative results (Sec. A.1), VBench and VBench-2.0 full results (Sec. A.2), Stage 1 training details (Sec. A.3), and Stage 2 training details (Sec. A.4). Sec. A.7 provides details about usage of LLMs in this project.

## A.1 QUALITATIVE RESULTS

We present qualitative comparisons in Figure A.1 across three scenarios involving human motion, object manipulation, and fluid dynamics. In each case, our method demonstrates superior temporal coherence and physical plausibility compared to existing baselines.

In the **first video**, depicting a man jumping, our method produces smooth and anatomically consistent motion. The subject maintains realistic posture and limb articulation throughout the sequence. In contrast, both CogVideoX and VideoREPA exhibit noticeable distortions, including unnatural body twisting and implausible joint movements. In the **second video**, where a person is eating pudding with a spoon, our model accurately captures the interaction: the spoon visibly scoops pudding, and the subject's mouth moves in coordination. The baselines fail to preserve this interaction — the person is either missing or the spoon disappears mid-sequence, breaking temporal and semantic consistency. In the **third video**, showing a man squeezing ketchup onto a plate, our method correctly models the accumulation of ketchup over time. The quantity on the plate increases as expected. Conversely, the baselines display erratic behavior: ketchup appears, disappears, and even flows back into the bottle, violating intuitive physical dynamics.

To facilitate further comparison, we include **eight video files** in the supplementary material. Each file contains a grid of six videos arranged in two rows and three columns. Each row corresponds to a different prompt, and each column shows the output from one of the three methods: CogVideoX, VideoREPA, and our MoAlign. This layout allows viewers to easily compare the outputs across methods for the same prompt and observe differences in motion consistency, semantic fidelity, and physical realism.

## A.2 VBENCH AND VBENCH-2.0 RESULTS

We present fine-grained results on the benchmarks in Tabs. A.1, A.2, A.3, A.4, and A.5.

**Notable decrease in dynamic degree and side-effect**. We observe that by applying MoAlign, dynamic degree drops most notably among all the other VBench metrics. The Dynamic Degree metric in VBench primarily measures the *magnitude* of pixel-space motion (e.g., optical-flow amplitude) rather than the physical plausibility of that motion. Upon closer inspection, we find that the higher dynamic degree scores achieved by the baseline CogVideoX largely stem from exaggerated or unstable motions — such as abrupt limb jumps, temporal jitter, or transient body parts — which inflate flow magnitude without corresponding to realistic dynamics.

On the other hand, MoAlign tends to reduce these unstable high-amplitude artifacts, leading to smoother and more physically grounded trajectories, which as expected lowers amplitude-based motion metrics. Importantly, note that this does not indicate a degenerate "low-motion" solution: physics-centric metrics such as VideoPhy and VideoPhy2 improve (e.g., +3.1 Joint), and the Joint metric is explicitly designed to penalize low-motion outputs. As shown in the Table 1, the Static Baseline, containing no motion, achieves a very high PC score but suffers a dramatic collapse in Joint, whereas MoAlign improves Joint over both the base model and the static case. This hints that MoAlign preserves meaningful motion while reducing mostly the nonphysical components. Qualitatively, the model continues to produce clear global motion (e.g., in dancing/running prompts), but with more realistic velocities and fewer nonphysical transitions.

## A.3 STAGE 1 TRAINING DETAILS

To learn a motion-centric representation disentangled from static appearance, we have a two-stage network to predict ground-truth optical flow from features extracted by a frozen VideoMAEv2 encoder. The encoder outputs token-level features of dimension 768, which are passed through a 3D convolutional compressor that reduces them to a 64-dimensional subspace. This compression network consists of two convolutional layers: the first is a temporal convolution with kernel size $(3, 1, 1)$ and padding $(1, 0, 0)$, which captures motion patterns across frames and maps the input to 256 channels. The second is a pointwise $(1 \times 1 \times 1)$ convolution that compresses the channel dimension to 64. Both layers are followed by SiLU activations. The input to this network is of shape $[B, 768, 24, H, W]$, and the output is $[B, 64, 24, H, W]$, representing our learned motion subspace optimized to retain dynamic structure while suppressing static semantics.

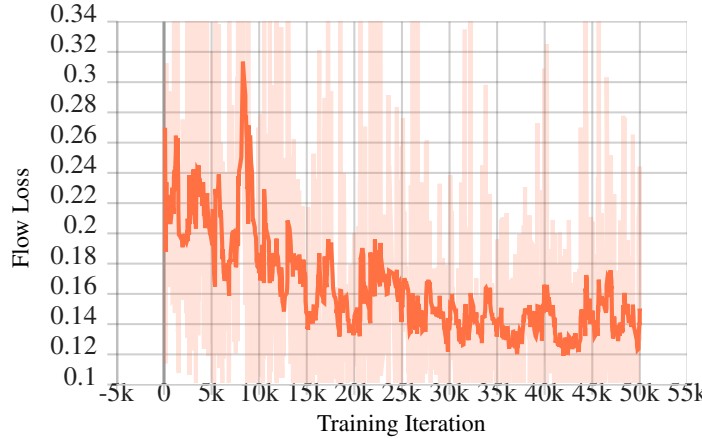

Figure A.2: Stage-1 motion projector head training convergence. The training converges at ∼30k iterations. The full 50k training takes ∼22 hours.

The compressed features are then fed into a flow prediction network designed to regress dense optical flow maps. This network begins with a 3D convolution that refines the temporal resolution from 24 to 23 using a kernel of $(2, 3, 3)$ and padding $(0, 1, 1)$, followed by a ReLU activation. It then applies two stages of spatial upsampling using transposed convolutions: the first upsamples by $2\times$ with a kernel of $(1, 4, 4)$ and stride $(1, 2, 2)$, mapping to 32 channels; the second upsamples again by $2\times$ to 16 channels. Each transposed convolution is followed by a ReLU. A final 3D convolution with kernel size 3 and padding 1 produces the flow vectors with 2 output channels. The resulting tensor is interpolated to a fixed shape of $[B, 2, 23, 128, 192]$ using trilinear interpolation, and then permuted to $[B, 23, 2, 128, 192]$ for compatibility with the ground-truth flow format.

The model is trained using L1 loss against RAFT-computed ground-truth flow, with all Video-MAEv2 weights kept frozen. We use the AdamW optimizer with a learning rate of $1 \times 10^{-4}$, $\beta_1 = 0.9$, $\beta_2 = 0.95$, and weight decay of $1 \times 10^{-3}$. Training is conducted for 50,000 iterations on four NVIDIA H100 GPUs (80GB VRAM each), with a batch size of 128 and mixed precision enabled via PyTorch AMP. Input videos are resized to $160 \times 240$ and truncated to 49 frames, with center cropping applied to ensure consistent input dimensions. Validation is performed every 1000 steps using a held-out set.

**Stage-1 training saturation**. Figure A.2 shows the training curve for learning the motion subspace from VideoMAE representations. We observe that the training saturates in around 30k iterations. The full training of 50k iterations for the first stage takes ∼22 hours. Note that the learned motion subspace projector is independent of the stage-2 training, the intrinsics of the diffusion model and its fine-tuning; thus the same motion sub-space projector once trained can be reused for various diffusion representation alignment episodes.

Table A.1: **VBench results** (part 1/2): consistency, motion, object-level metrics.

| Method | Subject Consistency | Background Consistency | Temporal Flickering | Motion Smoothness | Dynamic Degree | Aesthetic Quality | Imaging Quality | Object Class | Multiple Objects | Human Action |
|---|---|---|---|---|---|---|---|---|---|---|
| CogVideoX-2B | 92.9 | 94.7 | 97.1 | 97.6 | **70.3** | 62.9 | 63.2 | 86.8 | 66.8 | 97.2 |
| CogVideoX (FT) | 95.2 | 96.0 | 98.8 | 98.2 | 48.1 | 62.5 | 60.1 | 88.8 | 66.2 | 96.2 |
| VideoREPA | 95.7 | 96.3 | 98.9 | 98.2 | 44.4 | 63.2 | 61.1 | 88.2 | 71.1 | 96.2 |
| MoAlign (ours) | **95.8** | **96.4** | **99.0** | **98.4** | 42.2 | **64.5** | **64.5** | **89.6** | **75.2** | **98.4** |

Table A.2: **VBench results** (part 2/2): appearance, style, and overall scores.

| Method | Color | Spatial Relationship | Scene | Appearance Style | Temporal Style | Overall Consistency | Quality Score | Semantic Score | Total Score |
|---|---|---|---|---|---|---|---|---|---|
| CogVideoX-2B | 78.6 | 71.8 | 50.8 | 24.5 | **24.4** | **26.7** | 81.6 | 76.6 | 80.6 |
| CogVideoX (FT) | **84.4** | 69.6 | **52.4** | 24.4 | **24.4** | 26.3 | 81.1 | 77.1 | 80.3 |
| VideoREPA | 83.8 | 69.1 | 50.2 | **24.6** | **24.4** | 26.4 | 81.3 | 77.2 | 80.5 |
| MoAlign (ours) | 80.4 | **75.4** | 49.9 | 24.2 | 24.3 | 26.4 | **82.0** | **78.2** | **81.3** |

Table A.3: **VBench-2.0 results** (part 1/3).

| Method | Human Identity | Dynamic Spatial Relationship | Complex Landscape | Instance Preservation | Multi-View Consistency | Human Clothes | Dynamic Attribute | Complex Plot |
|---|---|---|---|---|---|---|---|---|
| CogVideoX-2B | 75.1 | 19.8 | 14.0 | 84.8 | **20.3** | 85.6 | **23.8** | 8.1 |
| CogVideoX (FT) | 79.7 | 18.8 | 13.3 | 91.8 | 8.7 | 88.1 | 17.6 | **9.2** |
| VideoREPA | **81.9** | 22.2 | 15.1 | 92.4 | 6.3 | 90.4 | 16.5 | 8.5 |
| MoAlign | 80.6 | **26.1** | **16.2** | **95.3** | 10.0 | **94.2** | 16.5 | 8.6 |

Table A.4: **VBench-2.0 results** (part 2/3).

| Method | Mechanics | Human Anatomy | Composition | Human Interaction | Motion Rationality | Material | Diversity | Motion Order Understanding |
|---|---|---|---|---|---|---|---|---|
| CogVideoX-2B | **64.2** | 82.6 | **55.6** | 60.3 | **35.6** | **68.1** | 50.0 | 10.1 |
| CogVideoX (FT) | 57.5 | 82.3 | 54.9 | 64.7 | 29.9 | 58.5 | **62.6** | 9.9 |
| VideoREPA | 55.6 | 84.0 | 55.6 | **66.3** | 30.5 | 61.5 | 58.2 | **10.6** |
| MoAlign | 57.9 | **85.3** | 53.3 | 64.7 | **35.6** | 64.6 | 52.3 | 8.9 |

Table A.5: **VBench-2.0 results** (part 3/3).

| Method | Camera Motion | Thermotics | Creativity Score | Commonsense Score | Controllability Score | Human Fidelity Score | Physics Score | Total Score |
|---|---|---|---|---|---|---|---|---|
| CogVideoX-2B | **49.7** | **63.0** | 52.8 | 60.2 | **26.6** | 81.1 | **53.9** | 54.9 |
| CogVideoX (FT) | 45.7 | 54.7 | **58.7** | 60.8 | 25.6 | 83.4 | 44.9 | 54.7 |
| VideoREPA | 42.3 | 57.1 | 56.9 | 61.4 | 25.9 | 85.4 | 45.1 | 55.0 |
| MoAlign | 38.9 | 62.8 | 52.8 | **65.5** | 25.7 | **86.7** | 48.8 | **55.9** |

## A.4 Stage 2 training details

In the second stage, we align the latent features of the CogVideoX diffusion model to the motion-centric subspace learned in Stage 1. This is achieved by introducing a lightweight projection network that maps internal representations from CogVideoX to the same $64$-dimensional space used for motion supervision. Specifically, we extract hidden states from the $18$th MM-DiT block of CogVideoX and pass them through a temporal-spatial projection head. This head first applies a temporal convolution stack consisting of a $(3, 1, 1)$ 3D convolution followed by a SiLU activation, and then a pointwise $(1 \times 1 \times 1)$ convolution with another SiLU activation. These layers reduce the input channel dimension from $1920$ to $256$ and then to $64$, preserving temporal structure while compressing appearance information.

After temporal processing, the features are interpolated along the time axis by a factor of $2$ using trilinear interpolation. The resulting tensor is then passed through a spatial downsampling layer to match with the spatial dimension of the videomae compressed features. This layer is a single $3 \times 3$ convolution with stride 3. This operation reduces the spatial resolution by approximately $3\times$, yielding a final output of shape that is compatible with the motion subspace.

During training, we freeze the VideoMAEv2 encoder and the motion compressor from Stage 1, and optimize only the CogVideoX transformer and the projection head. The training objective combines the standard denoising loss from the diffusion model with a soft relational alignment loss that matches the pairwise similarity structure of the projected features to those from the motion subspace. We use a temporal weighting scheme with $\tau = 10$ to emphasize long-range inter-frame consistency. The alignment loss is weighted by a factor of $0.5$ and added to the diffusion loss. Training is performed using AdamW with a learning rate of $2 \times 10^{-6}$, batch size 32, and mixed precision enabled. The model is trained for $4000$ steps on four NVIDIA H100 GPUs (80GB VRAM each), with validation conducted every $500$ steps using a held-out set of prompts.

**Stage-2 alignment depth.** As noted in Section 5.2, the choice of which CogVideoX block to align is a key factor for effective motion transfer. Prior alignment-based works (e.g., REPA, VideoREPA) similarly report that applying the alignment loss at a single, well-chosen mid-level layer yields the best performance, whereas spreading supervision across multiple blocks can overly constrain the denoising trajectory. In our case, we evaluated several transformer blocks of CogVideoX and found that aligning at the $18^{\text{th}}$ transformer block provides the strongest improvement in VideoPhy2 scores (see Tab. 5 in the main paper). We attribute the better performance of layers from the middle of the network to the internal dynamic of the denoising transformer. Namely, earlier layers primarily

encode low-level appearance features, while later blocks mainly refine high-frequency details and translate the representation to the output domain (since CogVideoX is not $x_0$-parametrized). While finding the theoretically justified algorithm for layer selection is an appealing research direction, in this work we opted for empirical comparison of several inner blocks.

## A.5 Experiments on WAN2.1 1.3B

To evaluate the effectiveness of our motion representation alignment method, we applied a similar alignment strategy to fine-tune WAN2.1 (1.3B), a widely used state-of-the-art efficient video diffusion model. Table A.6 presents the results. As shown, incorporating MoAlign fine-tuning improves all relevant metrics (SA, PC, and Joint score), though the gains are more moderate compared to those observed with CogVideoX.

Table A.6: Quantitative assessment of effectiveness of incorporating MoAlign on Wan2.1 (1.3B), as measured on VideoPhy2.

| Method | SA | PC | Joint |
|---|---|---|---|
| Wan-1.3B | 28.7 | 64.1 | 23.7 |
| Wan-1.3B + MoALign | **29.8** | **64.6** | **25.1** |

## A.6 Comparison with VideoJAM

To compare our MoAlign method with other approaches, we attempted to train the recently proposed VideoJAM (Chefer et al., 2025) modification of the video model. The training setup, base diffusion model, and dataset were identical to those used for Stage 2 of MoAlign (see Sec. 4.1). In our experiments, we found that under this training budget, VideoJAM remained far from convergence and produced unsatisfactory video outputs.

VideoJAM extends the original video generation model by simultaneously predicting the RGB visualization of the video's optical flow. This design introduces capabilities that were absent in the base model. The dynamics of training losses support this claim: at the start of training, the 'flow' component of the diffusion MSE loss is very high but gradually decreases, while the 'video' component is initially low — reflecting the model's prior video-generation pretraining — then rises, peaks, and slowly declines. This indicates that during the early phase of VideoJAM training, the video and flow objectives conflict, and the model requires substantial adaptation to align with the new joint objective. However, this adaptation is resource-intensive.

In the original work by Chefer et al. (2025), VideoJAM was trained on 3 million video clips (50K iterations on 32 GPUs) at a spatial resolution of $256 \times 256$. This far exceeds the requirements for MoAlign's Stage 2, which uses only 4K steps on 4 GPUs. We consider these findings strong evidence that our method offers better training efficiency for practitioners.

## A.7 Use of Large Language Models

We used Microsoft Copilot (a large language model) to aid in polishing the writing of this submission. The model was employed solely for improving clarity and readability; all ideas, technical content, and conclusions are our own.

