# OpenReview forum: "MoAlign: Motion-Centric Representation Alignment for Video Diffusion Models"
_ICLR.cc/2026/Conference — ICLR 2026 Poster_

### Official Review · Reviewer_evkb · 2025-10-31

**Soundness:** 3
**Presentation:** 3
**Contribution:** 3
**Rating:** 6
**Confidence:** 4

**Summary:**

MoAlign addresses a critical and timely problem of poor temporal coherence and physical implausability in videos generated by text-to-video diffusion models. The authors hypothesize that this failure stems from the entanglement of appearance and motion dynamics withon the model's internal representations. To address this, the authors provide a two stage fine-tuning framework. In the first stage, a motion centric feature subspace is learned by addition of an optical flow prediction head on top of a frozen video encoder. In the second stage, the internal features of a base text-to-video model are aligned with the distilled motion subspace using a soft relational alignment loss. This procedure is designed to internalize the motion knowledge disentangled from the appearance without adding overhead at inference time.

**Strengths:**

- The work addresses an important and timely problem. Although video models achieve generating visually plausible videos, most models lack the capability of generating physically plausible motion.
- The two stage framework is conceptually simple and well-explained. The idea of first learning a motion-specific latent subspace and then aligning the diffusion model's representations to that subspace is innovative. This decoupling of motion from appearance is a neat solution to force the model to internalize dynamics without confusion from static content.
- The paper's primary goal is to improve physical plausibility, and the results clearly demonstrate success. MoAlign consistently outperforms the base model, a fine-tuned baseline, and the re-implemented VideoREPA on the key "Physical Commonsense" (PC) and "Joint" metrics across both VideoPhy and VideoPhy2.
- The paper provides a wide range of experiments: four benchmarks + a user study + ablations. The ablation studies confirm the importance of each proposed component (motion-centric features and relational alignment). This thoroughness increases confidence in the approach. The paper also discusses examples of improved physical common sense (e.g., motion of objects following expected physics) which help illustrate the benefits.

**Weaknesses:**

- The paper's most significant flaw is its failure to cite, discuss, or compare against "VideoJAM: Joint Appearance-Motion Representations for Enhanced Motion Generation in Video Models" by Chefer et al. (2025). This omission is critical because VideoJAM addresses the exact same problem with a similar philosophy of incorporating an explicit motion signal, but through a fundamentally different mechanism. Both MoAlign and VideoJAM identify that the standard pixel-reconstruction objective is insufficient and biases models towards appearance at the expense of motion. Both propose to solve this by explicitly incorporating a motion-based objective during training.
- The improvements are mainly in motion consistency, as ensured by optical flow alignment. This addresses many physics issues (e.g., objects move more naturally). The paper doesn’t deeply analyze failure cases, but one can suspect that while MoAlign reduces obvious physics violations (like unnatural motion trajectories), it might still struggle with scenarios requiring understanding beyond motion (e.g., a generation might conserve motion but still violate gravity in subtle ways, etc.). A discussion of limitations would be useful.
- Although MoAlign is effective, parts of the approach are incremental. Prior work established that aligning diffusion features with a video model can inject knowledge (VideoREPA’s token relation loss is a direct precursor). MoAlign’s novelty mainly lies in isolating motion features via flow supervision. This is a sensible extension rather than a fundamentally new paradigm. The paper could do more to highlight what is truly new. For example, the idea of a learned motion projection head and the specific loss formulation could be emphasized as key innovations. Without clear differentiation, some readers may interpret MoAlign as a minor variant of VideoREPA (when in fact it has important differences). Ensuring the contributions are framed in contrast to related methods (especially REPA methods and VideoJAM) would strengthen the impact.

**Questions:**

- How does MoAlign compare with VideoJAM (Chefer et al. 2025) in approach and results? Both use optical flow to improve motion realism, VideoJAM adds a flow-prediction objective during training and uses an inner-guidance at inference.
- Your results consistently show a drop in Semantic Adherence (Table 2) and VBench-2.0 Physics score (Table 3) for all fine-tuned models. While you attribute this to the dataset, could you discuss the possibility that the alignment process itself, by strongly regularizing the model's feature space, inherently creates a trade-off that limits general capabilities? Was a "Pareto frontier" of plausibility vs. fidelity observed when tuning the alignment strength $\lambda$?
- You report that semantic fidelity is preserved, and user studies even noted improved realism. Can you comment on whether introducing the motion alignment had any side-effects on visual quality or diversity of generations?

---

> ### Author Response · Authors · 2025-11-25
>
> **W1/Q1 — Missing discussion and comparison with VideoJAM, which also introduces motion-based supervision to improve video generation.**
>
> The reviewer is right and we apologize for missing this closely related work. We have added an explicit discussion of **VideoJAM** (Chefer et al. 2025) in the introduction and related works section of the revised version of the manuscript. While both VideoJAM and MoAlign introduce motion-based supervision to counter appearance bias, the underlying mechanisms differ substantially. VideoJAM jointly predicts RGB optical flow during training and injects this prediction back into the denoising process via Inner-Guidance at inference. In contrast, MoAlign never predicts flow and requires no inference-time modification: we learn a motion-only representational subspace from a frozen `VideoMAEv2` encoder and align the diffusion model to this spatiotemporal structure. Motion is incorporated purely through representational alignment rather than through an auxiliary prediction head or guidance loop.
>
> Empirically, VideoJAM reports improvements primarily on VBench motion metrics, whereas MoAlign focuses on and evaluates physics-centric behavior (VideoPhy, VideoPhy2), which VideoJAM does not measure.
>
> Since VideoJAM is applied on top of a non-public video model and its implementation has not been released, reproducing their original results is not directly possible. However, based on the method description, we are currently implementing the core VideoJAM training objective and inference-time guidance on `CogVideoX-2B` to enable a fair comparison on an accessible backbone and on the same dataset, and we expect to report these results during the rebuttal period. We hope that the newly added discussion and forthcoming empirical comparison adequately address the reviewer’s raised point.
>
> ------------------------
>
> **W2 — Lack of discussion on failure cases and whether MoAlign still struggles with non-motion physics (e.g., gravity, multi-object interactions).**
>
>
> The reviewer is right: MoAlign does not solve all categories of physical reasoning. Our approach learns motion priors indirectly from a VideoMAEv2-based teacher whose motion subspace is shaped by real-world optical flow patterns. While this provides strong implicit physics (inertia, smooth trajectories, realistic velocities, contact-aware motion), it does not explicitly encode forces, material properties, or long-horizon causal effects. As a result, MoAlign can correct unrealistic motion trajectories but may still fail in scenarios requiring reasoning beyond motion alone, such as subtle gravity violations or complex multi-object interactions.
> However, adding such complex physics scenario based videos in the training dataset of both Stage-1 and Stage-2 might help address this limitation to a certain extent. We leave this exploration for future work.
> We have clarified this limitation and possible mitigation in the new “Limitations” section of the main paper.
>
> -----------------------
>
> **W3 — Clarify MoAlign’s novelty relative to REPA/VideoREPA and VideoJAM to avoid the impression that the method is merely incremental.**
>
> We thank the reviewer for this helpful suggestion. We agree that the paper can more clearly articulate its core novelty. In the revision, we explicitly rephrase our contribution to emphasize that MoAlign introduces a learned motion-only representational subspace, obtained through flow-supervised projection of a frozen VideoMAEv2 encoder, and aligns the diffusion model specifically to this motion-isolated space. This differs from prior alignment methods (e.g., REPA, VideoREPA, VideoJAM), which operate on joint appearance–motion representations or require predicting flow directly.
>
> We have revised the Contributions and Related Work sections to make this distinction precise: the key innovation of MoAlign is the separation of appearance and motion through a dedicated motion-only teacher, enabling physically grounded motion alignment without auxiliary prediction heads or inference-time guidance.

---

> ### Author Response · Authors · 2025-11-25
>
> **Q2 — Concern that the alignment loss ($\lambda$) may inherently trade off semantic fidelity for physical plausibility, potentially creating a Pareto frontier.**
>
> Indeed, as Tab. 2 shows, all finetuned models have lower overall Semantic Adherence (SA) score than the baseline. However, we would like to emphasize that the most noticeable drop occurs for the finetuned variant without any alignment, e.g., the FT modification. This setting corresponds to a  $\lambda = 0$ configuration, where no regularization toward motion-aware representations is applied, and we believe this is strong evidence that the reduction in SA can be mainly explained by the training data rather than the proposed MoAlign mechanism.
>
> In contrast, introducing MoAlign ($\lambda > 0$) substantially bridges the gap in SA compared to the baseline generator. In our experiments, $\lambda = 0.5$ provided the best balance between semantic fidelity and physics correctness, with both lower ($\lambda = 0.25$) and higher ($\lambda = 1.0$) values underperforming slightly. This suggests that $\lambda$ primarily controls the strength of motion-prior injection, but its impact is considerably smaller than that of the training data domain shift.
>
> Similar observations apply to the Physics score in Tab. 3. We have added clarifying remarks to the main text (see Sec. 5.1).
>
>
>
> | Method             | Overall SA    | Overall PC    |
> |--------------------|-------|--------|
> | *CogVideoX-2B*      | **49.8** | 23.9 |
> | *CogVideoX-2B (FT aka $\lambda = 0$)* | 44.9 | 34.1 |
> | **VideoREPA-2B** | 46.7 | 37.9 |
> | **MoAlign-2B (ours)** | 49.3 | **39.4** |
>
> * SA: Semantic Adherence; PC: Physical Commonsense
>
> --------------------
>
> **Q3 — Possible side-effects of MoAlign on visual quality and generative diversity.**
>
> Thank you for raising this important question. In addition to physics and Semantic Adherence (SA), we examined whether motion alignment introduces any unintended degradation in visual quality or generation diversity. On the original VBench, MoAlign maintains, or slightly improves, appearance-related dimensions such as aesthetic quality, subject consistency, and overall score relative to the fine-tuned baseline, suggesting that visual fidelity is not negatively affected.
>
> VBench also evaluates diversity through its “variation”–related metrics (inter-sample differences across multiple generations per prompt). We observed that MoAlign’s diversity scores remain comparable to those of the fine-tuned CogVideoX baseline, indicating that the method does not collapse the generation space or reduce variety. Our user study likewise did not report any loss of scene, action, or viewpoint diversity.
>
> Qualitatively, the main change introduced by MoAlign is a reduction of implausible high-frequency jitter and chaotic micro-motions, not a restriction of content diversity. The generated videos continue to cover a wide range of subjects and motions while displaying fewer nonphysical artifacts. We have updated the manuscript to reflect this insight (paragraph **Notable decrease in dynamic degree and side-effect** in appendix section A.2), and we thank the reviewer for highlighting it. This behavior precisely aligns with our goal of encouraging models to produce realistic motion rather than exaggerated, artifact-driven dynamics.

---

> > ### Author Response · Authors · 2025-12-02
> >
> > As discussed earlier, we conducted an experiment by training the VideoJAM modification of the CogVideoX-2B model using the same setup as Stage 2 of our MoAlign pipeline. Our results show that, under this training budget, VideoJAM remained far from convergence and produced unsatisfactory generations. We attribute this to the nature of the method: it introduces a new capability to the pretrained model. Although this capability is related to video generation, the additional objective is highly resource-intensive.
> >
> > In the original work, VideoJAM was trained on 3 million videos (50K iterations on 32 GPUs), whereas MoAlign requires only 4K steps on 4 GPUs. We consider these findings strong evidence that our method offers superior training efficiency for practitioners.
> >
> > We have updated the manuscript to include details of this experiment in Sec. A6. We thank the reviewer for this suggestion, which helped us further investigate the training efficiency of our proposed approach.

---

> ### Comment · Reviewer_evkb · 2025-11-27
>
> The rebuttal and revisions have addressed my main concerns.
>
> The authors now clearly position MoAlign with respect to VideoJAM and prior alignment methods, emphasizing the motion-only subspace and representational alignment without inference-time guidance. They have also added a limitations section, accurately clarifying that MoAlign mainly improves motion statistics and physics-related behavior but does not solve all forms of physical reasoning. The novelty relative to prior REPA-based methods is also better articulated.
>
> Regarding the trade-offs, the additional analysis suggests that the Semantic Adherence drop stems from the fine-tuning data rather than the alignment itself. I accept the finding that an intermediate alignment strength yields a reasonable balance and that visual quality and diversity are largely preserved.
>
> Overall, I keep my score, but following the rebuttal, I am more comfortable with the paper's acceptance.

---

### Official Review · Reviewer_mRuk · 2025-10-31

**Soundness:** 3
**Presentation:** 3
**Contribution:** 3
**Rating:** 6
**Confidence:** 4

**Summary:**

This paper proposes MoAlign, to improve the physical plausibility and temporal coherence of T2V diffusion models. This framework first creates a motion-centric feature with a frozen video encoder to predict optical flow, then the diffusion model's features are aligned to this motion subspace with alignment loss. The results show improvements in physical commonsense benchmarks over baselines, without quality drops.

**Strengths:**

1. The paper identifies a key limitation of existing representation between appearance and motion. The proposed two-stage approach is a novel and intuitive solution.

2. Using optical flow as an explicit signal to capture motion-centric representation space is a logical and strong choice into the diffusion model.

3. The paper validates its approach using a wide range of benchmarks.

**Weaknesses:**

1. Although the effectiveness of proposed method is validated by physical-centric benchmarks, it is demonstrated exclusively on CogVideoX-2B. It is unclear if this motion alignment approach would yield similar benefits for other SoTA video diffusion architectures, which may have different internal representations.

2. The claim of improved plausibility is accompanied by a drop in metrics related to motion dynamism. Specifically, MoAlign shows a large reduction in VBench's 'Dynamic Degree' (70.3 $\rightarrow$ 42.2) and VBench-2.0's 'Dynamic Attribute' (23.8 $\rightarrow$ 16.5). Is the model learning to be safer by simply reducing overall motion to avoid making physical mistakes?

3. The model also shows a notable drop in VBench-2.0's 'Physics Score' (53.9 $\rightarrow$ 48.8). The authors attribute this (L410) to a lack of training samples for phenomena like thermotics and materials. I wonder if the model can easily incorporate these missing physics domains via fine-tuning (e.g., SFT or MoAlign method on a few examples) or if it would require a complete retraining.

**Questions:**

Refer to weaknesses.

---

> ### Author Response · Authors · 2025-11-25
>
> **W1 — Unclear whether MoAlign generalizes beyond CogVideoX-2B to other video diffusion architectures.**
>
>
> Thank you for raising this question about generality across architectures. We have applied MoAlign to a second and substantially different backbone, **Wan-1.3B** (Wan et al. 2025), whose architecture (non-MMDiT, global spatiotemporal attention) does not exhibit the localized temporal attention patterns present in CogVideoX (MMDiT-based). Despite this structural difference, MoAlign still provides consistent improvements on VideoPhy2, as shown below:
>
> | Method        | SA   | PC   | Joint |
> |---------------|------|------|--------|
> | Wan-1.3B      | 28.7 | 64.1 | 23.7   |
> | Wan-1.3B + MoAlign | **29.8** | **64.6** | **25.1** |
> * SA: Semantic Adherence; PC: Physical Commonsense
>
> While the improvements are more modest than those on CogVideoX, this behavior is consistent with observations reported in prior work. In particular, the authors of DeT publicly note that their motion-enhancement method fails to run or diverges entirely on WAN’s architecture due to its different temporal-attention structure (see Issue #2 on the official third-party repository: https://github.com/Shi-qingyu/DeT/issues/2). To our knowledge, most recent motion-centric approaches (including DeT and VideoREPA variants) are evaluated exclusively on CogVideoX or HunyuanVideo for this reason.
> In contrast, MoAlign does not introduce architectural assumptions such as temporal kernels or local convolutions; it only aligns internal representations. This makes the method stable on WAN and sufficient to obtain modest but consistent gains where several prior motion-enhancement approaches fail.
> We have created a new subsection (**A.5 Experiments on WAN2.1 1.3B**) to cover the new findings, and have given a reference to it in the main text.

---

> ### Author Response · Authors · 2025-11-25
>
> **W2 — Concern that MoAlign reduces overall motion amplitude rather than improving motion plausibility.**
>
>
> Thank you for this insightful observation. The Dynamic Degree metric in VBench primarily measures the magnitude of pixel-space motion (e.g., optical-flow amplitude) rather than the physical plausibility of that motion. Upon closer inspection, we find that the higher dynamic degree scores achieved by the baseline CogVideoX largely stem from exaggerated or unstable motions, such as abrupt limb jumps, temporal jitter, or transient body parts, which inflate flow magnitude without corresponding to realistic dynamics.
>
> MoAlign tends to reduce these unstable high-amplitude artifacts, leading to smoother and more physically grounded trajectories, which as expected lowers amplitude-based motion metrics. Importantly, please note that this does not indicate a degenerate “low-motion” solution: physics-centric metrics such as VideoPhy and VideoPhy2 improve (e.g., +3.1 Joint), and the Joint metric in VideoPhy2 is explicitly designed to penalize low-motion outputs. As illustrated in the table below, the Static Baseline, containing no motion, achieves a very high PC score but suffers a dramatic collapse in Joint, whereas MoAlign improves Joint over both the base model and the static case. This hints that MoAlign preserves meaningful motion while reducing mostly the nonphysical components. Qualitatively, the model continues to produce clear global motion (e.g., in dancing/running prompts), but with more realistic velocities and fewer nonphysical transitions.
>
>
> | Method             | SA    | PC    | Joint |
> |--------------------|-------|-------|--------|
> | *CogVideoX-2B*      | 27.1 | 64.5 | 22.3   |
> | **Static baseline** | 15.6 | **91.0** | 15.1   |
> | **MoAlign-2B (ours)** | **28.8** | 75.0 | **24.9** |
> * SA: Semantic Adherence; PC: Physical Commonsense
>
> We have updated the manuscript to reflect this insight (paragraph **Notable decrease in dynamic degree and side-effect** in appendix section A.2), and we thank the reviewer for highlighting it. This behavior precisely aligns with our goal of encouraging models to produce realistic motion rather than exaggerated, artifact-driven dynamics.
>
> --------------------------------
>
> **W3 — Drop in VBench-2.0 Physics Score and whether missing physics domains can be incorporated without full retraining.**
>
>
> We appreciate the reviewer’s careful observation. The drop in the VBench-2.0 Physics score arises because this metric spans physics domains (e.g., thermotics, material deformation, liquids, granular media) that are absent across both components of our pipeline: the Stage-1 motion-teacher training data and the diffusion model’s Stage-2 finetuning data. In contrast, our training data for MoAlign is more suitable for dynamic, rigid-body, and action-physics categories, which is where VideoPhy and VideoPhy2 show improvements.
>
> Importantly, addressing these additional physics domains does not require retraining VideoMAE or CogVideoX from scratch. If videos containing such phenomena become available, MoAlign can incorporate them by retraining only Stage-1, which is a lightweight and fully decoupled step (≈22 hours for 50k iterations on 4×H100). The diffusion model can then absorb these expanded motion priors through our standard Stage-2 alignment (≈24 hours for 4k iterations on 4×H100). We have updated the manuscript to clarify this distinction and discuss this limitation explicitly in the new “Limitations” section in the main paper.

---

### Official Review · Reviewer_DP4f · 2025-11-01

**Soundness:** 3
**Presentation:** 3
**Contribution:** 3
**Rating:** 6
**Confidence:** 3

**Summary:**

This paper proposes MoAlign, a motion-centric representation alignment framework for video diffusion models. The method learns a motion-specific subspace from a pretrained video encoder by supervising its projection to predict optical flow, and then aligns diffusion features to this subspace through a soft relational alignment loss.

**Strengths:**

**1) Relevance and Timeliness**

Motion-centric alignment addresses a timely limitation in large video diffusion models, namely the lack of explicit motion understanding, and the approach fits well within the ongoing trend of representation-based fine-tuning.

**2) Clear Methodology**

The two-stage design (motion feature learning → diffusion feature alignment) is logically presented and easy to follow, providing a clean conceptual link between motion representation learning and diffusion adaptation.

**3) Comprehensive Evaluation**

The paper conducts extensive experiments on diverse benchmarks and includes both quantitative metrics and human preference studies, demonstrating consistent improvements in motion and physical realism.

**Weaknesses:**

**1) Literature Coverage**

While the introduction claims that prior alignment-based methods mostly focus on visual semantics rather than true motion dynamics, the paper overlooks several recent studies that explicitly target motion-aligned or dynamics-centric representations (e.g.[1], [2]). These works similarly aim to internalize motion dynamics rather than relying on appearance cues. Clarifying how the proposed framework surpasses such motion-aware alignment approaches would help position this work more precisely within the current landscape and better substantiate its claimed novelty and advantage.

**2) Efficiency and Practical Overhead**

Although the paper emphasizes that the proposed framework internalizes motion understanding without external simulators or inference-time conditioning, the Stage 1 training involves over 50 K iterations using multiple H100 GPUs. This introduces substantial computational overhead and weakens the claim of simplicity or efficiency. It would be informative to discuss whether this stage truly needs such heavy training to be effective. For instance, how does performance change if the motion projector is trained with fewer iterations or a lighter backbone?

**3) Limited Discussion on Motion Representation Across Layers**

Prior alignment-based works (e.g., REPA, VideoREPA) have shown that the choice of layer for alignment is a critical design factor, often focusing on mid or multiple layers and providing useful insights into how different layers contribute to the denoising process and what types of information they encode. In this work, however, the alignment is applied to a single layer without further discussion, and despite the paper’s focus on learning motion-related representations, it lacks sufficient analysis or discussion of how the aligned layer captures motion information. It would strengthen the paper to elaborate on what insights this approach provides regarding how motion-related representations are distributed across layers, as this understanding could offer a deeper contribution to motion-centric representation learning.

[1] *Track4Gen: Teaching Video Diffusion Models to Track Points Improves Video Generation*
[2] ​​*VideoJAM: Joint Appearance-Motion Representations for Enhanced Motion Generation in Video Models*

**Questions:**

See the weakness

---

> ### Author Response · Authors · 2025-11-25
>
> **W1 — Insufficient discussion of recent motion-alignment works (Track4Gen, VideoJAM) and unclear novelty positioning.**
>
> We thank the reviewer for pointing out these highly relevant concurrent works. **Track4Gen** (Jeong et al. 2025) and **VideoJAM** (Chefer et al. 2025) indeed introduce motion supervision. Track4Gen operates in an image-to-video setting and uses optical-flow-derived point trajectories to enforce local correspondence consistency at a single UNet block; it does not address or evaluate global motion dynamics or physical plausibility as we do. VideoJAM jointly predicts RGB optical flow and appearance and uses inference-time inner-guidance, meaning motion is injected through a learned auxiliary output that modifies the sampling trajectory. In contrast, our method does not predict flow nor require any inference-time modifications: we learn a motion-only subspace from a frozen `VideoMAE` encoder and align the diffusion transformer to its spatio-temporal relational geometry, enabling the model to internalize motion priors without altering the generation interface.
>
> Critically, neither Track4Gen nor VideoJAM evaluate physical plausibility on third-party physics benchmarks, whereas MoAlign is explicitly validated on such benchmarks like VideoPhy and VideoPhy2, which assess dynamics such as rigid-body behavior, contact interactions, and material responses: metrics that go beyond generic VBench motion smoothness. We have added citations and explicit discussion of these differences in the revised Related Work section.
>
> Since VideoJAM is applied on top of a non-public video model and its implementation has not been released, reproducing their original results is not directly possible. However, based on the method description, we are implementing the core VideoJAM training objective and inference-time guidance on `CogVideoX-2B` to enable a fair comparison on an accessible backbone, and we expect to report these results during the rebuttal period.

---

> > ### Author Response · Authors · 2025-12-02
> >
> > As discussed earlier, we conducted an experiment by training the VideoJAM modification of the CogVideoX-2B model using the same setup as Stage 2 of our MoAlign pipeline. Our results show that, under this training budget, VideoJAM remained far from convergence and produced unsatisfactory generations. We attribute this to the nature of the method: it introduces a new capability to the pretrained model. Although this capability is related to video generation, the additional objective is highly resource-intensive.
> >
> > In the original work, VideoJAM was trained on 3 million videos (50K iterations on 32 GPUs), whereas MoAlign requires only 4K steps on 4 GPUs. We consider these findings strong evidence that our method offers superior training efficiency for practitioners.
> >
> > We have updated the manuscript to include details of this experiment in Sec. A6. We thank the reviewer for this suggestion, which helped us further investigate the training efficiency of our proposed approach.

---

> ### Author Response · Authors · 2025-11-25
>
> **W2 — Concern about Stage-1 training cost and whether 50k iterations on multiple H100s undermine the efficiency claim.**
>
> We thank the reviewer for sharing this concern. Indeed Stage-1 introduces additional compute. Our initial choice of 50k iterations followed standard practice in VideoMAE-style pretraining and was selected conservatively to ensure stable convergence. In the supplementary material, we now include a convergence curve (Figure A.2) showing that the flow-supervised motion projector begins to saturate at around 30k iterations (~14 hours), while 50k iterations require ~22 hours in total on 4×H100 GPUs.
>
> Please note that Stage-1 is a fully decoupled, one-time offline step: it is trained independently of any diffusion model (no joint optimization, no shared gradients), and its learned motion subspace can be reused across different experiments, datasets, and even entirely different video diffusion architectures. This amortizes the cost and avoids repeated retraining. Stage-2 and inference incur no additional overhead, preserving the simplicity of the generation pipeline. We have clarified this in the revised manuscript.
>
> --------------------------------
>
> **W3 — Lack of analysis on why alignment is applied to a single mid-layer and how motion representations are distributed across layers.**
>
>
> We agree that the choice of alignment layer is important. Prior alignment works (REPA, VideoREPA) similarly apply their loss at a single, well-chosen mid-depth layer rather than across all blocks. For example, the official REPA implementation aligns to encoder-depth 8, and VideoREPA’s analysis identifies alignment depth ≈18 as optimal for CogVideoX, with deeper layers becoming overly constrained when regularized too strongly. These observations motivate focusing the alignment where the model naturally encodes mid-level relational structure rather than imposing supervision uniformly across the network.
>
> Our results show the same pattern. As summarized in Table 5 in the main paper (reproduced below), aligning MoAlign at layer 18 yields the best VideoPhy2 performance, while both earlier and later layers lead to lower physical commonsense (PC) scores. In our internal experiments, spreading the loss across multiple layers slightly degraded VideoPhy2 metrics, suggesting that injecting a motion-only prior too broadly over-regularizes the denoising trajectory and harms the appearance–physics balance. We have added this discussion in the ablation section of the main paper.
>
> | Layer | SA   | PC   | Joint |
> |-------|------|------|--------|
> | 10    | 27.3 | 71.9 | 23.2   |
> | 12    | 27.6 | 71.7 | 22.9   |
> | 14    | 27.1 | 71.6 | 23.1   |
> | 16    | 28.2 | 73.9 | 24.2   |
> | **18** | **28.8** | **75.0** | **24.9** |
> | 20    | 27.1 | 73.2 | 23.8   |
> | 22    | 27.6 | 72.8 | 23.4   |
> * SA: Semantic Adherence; PC: Physical Commonsense
>
> We attribute the better performance of layers from the middle of the network to the internal dynamic of the denoising transformer.
> In detail, earlier layers primarily encode low-level appearance features, while later blocks mainly refine high-frequency details and translate the representation to the output domain (since CogVideoX is not $x_0$-parametrized).
> While finding the theoretically justified algorithm for layer selection is an appealing research direction, in this work we opted for empirical comparison of several inner blocks.

---

### Author Response · Authors · 2025-11-25

We sincerely thank all reviewers for their time and constructive feedback. The overall reception of *MoAlign* has been encouraging, with ratings of 6/6/6 reflecting recognition of its relevance, clarity, and empirical rigor. Reviewers noted that the work addresses a timely challenge in text-to-video diffusion models, namely the lack of explicit motion understanding and physical plausibility, and found the proposed two-stage framework to be clear and well-motivated. The idea of learning a motion-focused subspace via optical flow supervision and aligning diffusion features to this subspace was considered a sound and intuitive approach. Specific positive aspects highlighted across reviews include:

- **Relevance and Impact**: The problem of motion-appearance entanglement and improving physical realism was seen as one of "relevance and timeliness" (DP4f) and “important and timely” (evkb).
- **Design and Approach**: The two-stage structure and use of optical flow as an explicit signal were regarded as “logically presented” (DP4f), “novel and intuitive” (mRuk) and “innovative and neat” (evkb).
- **Empirical Validation**: The breadth of evaluation, including multiple benchmarks, user studies, and ablations, was appreciated (mRuk,DP4f, evkb).
- **Practical Advantage**: Moalign’s design internalizes the motion knowledge disentangled from the appearance without adding overhead at inference time (evkb).
- **Conceptual Link**: The framework provides a clear and logical connection between motion representation learning and diffusion adaptation, which was considered conceptually strong (DP4f).

We are grateful for these constructive suggestions, which have helped us refine the positioning and clarity of our contributions. All questions and suggestions for improvements including broader comparisons to recent motion-aware approaches (e.g., VideoJAM), efficiency considerations for Stage 1 training, and deeper analysis of alignment across layers and trade-offs between plausibility and dynamism have been discussed in detail in the individual responses.

---

### Meta-Review · Area_Chair_dnJA · 2026-01-07

**Summary:**

A major recurring critique is that the paper does not sufficiently position itself against recent motion-aware methods, especially VideoJAM and also works like Track4Gen. Reviewers argue the omission is important because these approaches share a similar motivation (inject explicit motion signals) but differ mechanistically. Several reviewers felt that without explicit comparison and clearer differentiation, MoAlign may appear as an incremental extension of VideoREPA (novelty concentrated mainly in the flow-supervised motion subspace).

Another concern is the compute and practical overhead. While MoAlign avoids inference-time conditioning or simulators, reviewers question the practical cost of Stage-1 training (e.g., 50k iterations on multiple H100s). They ask whether Stage 1 must be that expensive, how performance scales with fewer iterations / lighter models, and whether the “simplicity/efficiency” framing should be moderated or better justified.

MoAlign reduces motion-related metrics (notably VBench Dynamic Degree and VBench-2.0 Dynamic Attribute). Reviewers ask whether the model improves physical plausibility partly by reducing overall motion amplitude, i.e., avoiding errors by being conservative. They want stronger analysis that the gains are not explained by motion suppression, and more discussion of when MoAlign may reduce desirable dynamics.

One reviewer highlights that results are primarily shown on CogVideoX-2B, and it is unclear whether motion-centric alignment transfers broadly to other architectures. Even if the appendix contains additional backbone results, reviewers request clearer evidence and discussion in the main narrative.

Reviewers note drops in some physics-related aggregate scores (e.g., VBench-2.0 Physics), with the authors attributing this to missing domains like thermotics/materials in training data. They request a clearer discussion of whether missing domains can be incorporated via additional fine-tuning/data, and more explicit failure modes (beyond motion trajectory issues).

**Reviewer Concerns:**

Concerns Likely Addressed in the Authors’ Response
- Training cost / efficiency claim. Authors clarify Stage-1 is decoupled and reusable, provide convergence evidence (saturates ~30k steps), and argue cost is amortized across backbones/experiments; Stage-2/inference has no overhead. The rebuttal reframes cost as one-time offline and supports with concrete numbers.
- Motion suppression concern. Authors argue VBench “Dynamic Degree” measures flow magnitude, often inflated by artifacts; show that static baseline collapses Joint, while MoAlign improves Joint, suggesting it’s not merely “no motion.”
- Generalization beyond CogVideoX. Authors add results on Wan-1.3B, showing consistent (though smaller) improvements, and argue representation alignment is less architecture-dependent.

Concerns Partially Addressed
- Missing/unclear positioning vs. VideoJAM / Track4Gen and novelty framing. Authors now explicitly contrast MoAlign with Track4Gen (I2V + local correspondence) and VideoJAM (joint flow+RGB prediction + inference-time inner-guidance), emphasizing MoAlign’s motion-only subspace and no inference-time modification / no flow prediction. Clear conceptual comparison is provided, but empirical comparison with VideoJAM (and other concurrent motion-supervised approaches) is not yet shown (only planned).
- VBench-2.0 Physics score drop. Authors attribute drop to data coverage, explain that adding domains requires retraining Stage-1 + Stage-2, not full retraining, and add a limitations section.
- Failure cases / limitations. A limitations section is added, acknowledging MoAlign mainly improves motion statistics rather than full physical reasoning.

Concerns Still Likely Outstanding
- Incremental concern relative to VideoREPA. Authors provide improved framing (motion-only subspace is the key novelty), plus ablations support benefit. But for some reviewers, novelty is a conceptual bar; rebuttal helps, but may not fully change their underlying prior that it’s a “sensible extension” rather than a new paradigm.

**Reviewer Scores:**

Reviewer DP4f

Original rating: 6 (marginal accept)

Likely change: Remain at 6 (with a more positive acceptance stance)

Reason: The rebuttal substantially improves related-work positioning (Track4Gen/VideoJAM), clarifies Stage-1 cost amortization with concrete convergence evidence, and strengthens the rationale for single-layer alignment via a layer sweep. However, the most load-bearing gap—an empirical baseline comparison to VideoJAM—remains pending (promised rather than shown), and the reviewer’s original “borderline / wouldn’t mind rejection” tone suggests the score is unlikely to increase without that missing experiment.

Reviewer mRuk

Original rating: 6 (marginally above acceptance threshold)

Likely change: increase to 8 or remain 6

Reason: The authors directly address the three key concerns: (i) provide cross-backbone evidence on Wan-1.3B supporting generality, (ii) give a compelling argument (with a static-baseline sanity check) that reduced “Dynamic Degree” reflects removal of nonphysical high-amplitude artifacts rather than a degenerate low-motion solution, and (iii) clarify that missing physics domains can be incorporated by retraining only lightweight Stage-1 plus standard Stage-2, not full retraining. These points are likely sufficient to move a marginal accept toward weak accept, though the modest WAN gains and the remaining physics-coverage gap could keep the reviewer at 6.

Reviewer evkb

Original rating: 6 (marginally above acceptance threshold)

Likely change: remain at 6

Reason: The reviewer explicitly states the rebuttal and revisions addressed their main concerns (VideoJAM positioning, limitations, novelty clarification, and trade-off discussion), and that they are “more comfortable with acceptance,” but they also explicitly indicate they will keep their score.

---

### Decision · Program_Chairs · 2026-01-26

Accept (Poster)